# The magnitude and effect of work-life imbalance on cognition and affective range among the non-western population: A study from Muscat

**Samir Al-Adawi**[1]*, **Mohamad Alameddine**[2,3], **Muna Al-Saadoon**[4], **Amal A. Al Balushi**[5], **Moon Fai Chan**[6], **Karen Bou-Karroum**[7], **Hamad Al-Kindy**[8,9], **Saud M. Al-Harthi**[9]

1 Department of Behavioral Medicine, College of Medicine and Health Sciences, Sultan Qaboos University, Muscat, Oman, 2 College of Health Sciences, University of Sharjah, Sharjah, United Arab Emirates, 3 College of Medicine, Mohammed Bin Rashid University of Medicine and Health Sciences, Dubai Health Care City, Dubai, United Arab Emirates, 4 Department of Child Health, College of Medicine and Health Sciences, Sultan Qaboos University, Muscat, Oman, 5 Higher Medical Committee, Ministry of Health, Muscat, Oman, 6 Department of Family Medicine and Public Health, College of Medicine and Health Sciences, Sultan Qaboos University, Muscat, Oman, 7 Faculty of Health Sciences, Department of Health Management and Policy, American University of Beirut, Beirut, Lebanon, 8 Compensation Board, Directorate General of Khoula Hospital, Ministry of Health, Muscat, Sultanate of Oman, 9 Muscat Directorate of General Health Services, Ministry of Health, Muscat, Oman

* adawi@squ.edu.om, samir.al-adawi@fulbrightmail.org

**Data Availability Statement:** The data that support the findings of this study are available on request from the corresponding author. The data are not

## Abstract

The temporal relationship between work-life balance/imbalance, occupational burnout, and poor mental health outcomes have been widely explored. Little has been forthcoming on cognitive functioning among those with work-life imbalance. This study aimed to explore the rate of work-life imbalance and the variation in neuropsychological functioning. The relationship between affective ranges (anxiety and depressive symptoms) and work-life balance was also explored. The target population in this study are Omani nationals who were referred for psychometric evaluation. The study employs neuropsychology measures tapping into attention and concentration, learning and remembering, processing speed, and executive functioning. Subjective measures of cognitive decline and affective ranges were also explored. A total of 168 subjects (75.3% of the responders) were considered to be at a work-life imbalance. Multivariate analysis showed that demographic and neuropsychological variables were significant risk factors for work-life imbalance including age and the presence of anxiety disorder. Furthermore, participants indicating work-life imbalance were more likely to report cognitive decline on indices of attention, concentration, learning, and remembering. This study reveals that individuals with work-life imbalance might dent the integrity of cognition including attention and concentration, learning and remembering, executive functioning, and endorsed case-ness for anxiety.

publicly available due to privacy or ethical restrictions. Sharing this data is restricted by Omani data protection laws. The data used in this study cannot be deposited in publicly accessible archives. Access to data can be obtained by application to the Postgraduate Studies & Research (http://www.squ.edu.om/dor), College of Medicine and Health Sciences, Sultan Qaboos University. mo.ude.uqs@rsgpdem.

**Funding:** This study was funded by the MBRU-AlMahmeed Award number AIM1824. The Award was received by SAA. The funders had no role in study design, data collection and analysis, decision to publish, or preparation of the manuscript.

**Competing interests:** The authors have declared that no competing interests exist.

## Introduction

Modern industrial economies that once only dominated landscapes in Europe, North America, Japan, and Australia are now increasingly encroaching on societies in transition, often labeled "emerging economies" or "developing countries" [1]. As a result, urbanization and dependence on the modern cash economy have become a rule rather than an exception. Amid such changes, there is growing evidence to suggest that occupational stress or burnout once reported in industrialized countries are increasingly becoming common in many such societies in transition [2]. The Chinese sage, Confucius, who lived in the 6th century, is credited to have said: "choose a job you love, and you will never have to work a day in your life". Fast-forward to the 21st century and the ideal world of Confucius is teaching that appears untenable to most people in the labor force. The World Health Organization has recently labeled occupational burnout as a syndrome linked to long-term, unresolved, work-related stress [3].

While the relationship between poor coping at the workplace and its temporal relationship with temperament and psychiatric distress has been the subject of empirical scrutiny in the last decades [4–6], another line of research, often coming under the umbrella of 'work-life imbalance', 'work-family conflict', 'work-nonwork interface' or its counterpart, 'work-life imbalance', 'work-life conflict', is becoming 'mainstream' [7]. These terms denote "lack of time for personal affairs such as mundane family obligations, regular exercise and other forms of social life' because one is engaged with work [8].

The adverse outcomes of burnout have been outline in literature which include poor coping, absenteeism, illness, staff conflict, and substance abuse [9]. Studies have suggested that occupational burnout is significantly associated with work-life imbalance, although it is still to be figured out which one causes the other.

Work-life balance is defined as the failure of the employee to juggle, without undue distress, different spheres of life [10]. Some theoretical papers have indicated that modernity has put a strain on work-life balance [11]. The work-life imbalance may partly stem from the influx of technology that is currently encroaching on multiple spheres of life [11]. Individuals are, therefore, obligated to attend to work even outside the occupational setting. Some studies have suggested that changing gender roles and the required length of the work hours per week has also marred work-life balance [12, 13]. The work-life imbalance is associated with increased sick leaves [14], high turnover [15], as well as increased risk factors for many types of refractory physical and mental conditions [16].

While ample studies have examined the psychosocial impact of work-life balance, to our knowledge, there is a dearth of studies that have examined the association between work-life imbalance and the cognitive functioning or integrity of neuropsychological functioning (e.g attention and concertation, registration and recall, executive functioning, and processing speed). While such a study may fall short of establishing causality, it has the potential to lay the groundwork by unearthing significant associations.

Neuroscience literature is replete with instruments capable of systematically exploring the neural substrate of information processing in the brain. Applying neuropsychological tools on the trajectories of work-life imbalance has the potential to shed light on how activities of daily living shape recognizable traces at the neural level, if any [17].

In the industrialized countries, studies have shown work-life imbalance is associated with cognitive decline as measured by neuropsychological tools [18, 19]. The following assumption has also been put forth: stressors at the workplace and accompanying work-life imbalance tend to trigger a cascade of adverse neurochemical changes which in turn impact brain regions that are critically involved in higher functioning including the hippocampus and the prefrontal cortex [20–22]. Some studies have suggested that prolonged work-life imbalance tends to

elevate neuro-hormonal activities, which in turn trigger reduction of volume in the brain region critically involved in cognition [20–22]. Most of these studies have emanated from the Euro-American regions.

Studies have suggested that the prolonged stress or specifically occupational stress or the presence of poor mental health outcome tend to trigger cognitive impairment [18, 19, 23]. One hypothesis suggest that impaired thinking process such as attentional bias in anxiety symptom and negative bias in depressive symptom are likely to dent the integrity of cognitive functioning [24]. In support of such view, both anxiety and depressive symptoms with impaired cognition tend to have attenuation of brain activity in critical brain areas [25].

This study is a rare attempt to examine such relationships in a developing context, specifically at the Sultanate of Oman, a country located in the southern tip of the Arabia Peninsula. The aims of this study were to (i) explore the rate of work-life imbalance among Omanis with poor adjustment at work, (ii) compare demographic, reasoning ability, and neuropsychological factors between people has indicated work-life balance and work-life imbalance, and (iii) explore the factors that contributing to the work-life imbalance.

## Methods

### Setting

The study was conducted from April 2018 to May 2019 at a tertiary care center in Muscat, the national capital. Oman has a free universal healthcare system centralized and compartmentalized for its citizens [26]. The first point of contact for healthcare consultation in Oman is primary care, and if needed, the client is referred by a generalist to secondary or tertiary care centers distributed across different parts of Oman [27]. The participants of this study were those referred for psychometric evaluation to the present unit, the Department of Behavioural Medicine at a tertiary care center. The present study participants constitute those who have been referred for such evaluation of their fitness to work and meeting the criteria for burnout or life-management difficulty as defined by the International Statistical Classification of Diseases and Related Health Problems (ICD-10) [28]. According to the Oman labor law, employees who have been underperforming at their workplace or accruing many sick leaves are referred for psychometric evaluation to determine their occupational competence/fitness to work, and whether they are entitled to medical boarding or to transfer to less taxing workload. The exclusion criteria included job-seekers, retired individuals and individuals with a pervasive and persistent psychiatric, medical and neurological condition. Similarly, participants working in shifts at the time were also excluded under the pretext that shift-workers tend to present significant confounder in neuropsychological measures [29, 30].

### Power analysis

A logistic regression model was used to estimate the required samples for this study. Based on previous studies [20–22], if we expected the prevalence rate of having the work-life balance to be 25%, the expected coefficient of anxiety scores on work-life balance ranged from -0.60 to -0.50. The minimum required sample size was determined to be 168 individuals; this would help achieve 80% power at a 5% level of significance [31].

### Outcome measures

The outcome measures were those that solicit the presence of work-life balance, intellectual capacity and neuropsychological functioning. Mood states–anxiety and depression—were also solicited.

**Work-life balance.** While the trajectories of work-life balance have been well-recognized in the literature [32], there is a dearth of well-established instruments to measure it. To solicit the presence of work-life balance, the present study employed a brief questionnaire examining whether the relationship between work and non-work life is optimal and whether the resources (such as time and energy) left for non-work life are adequate. This questionnaire has been used elsewhere [33]. The participant who answered affirmatively on the two aforementioned dimensions were considered to have a work-life balance. All other answers were considered to be indicative of work-life imbalance.

**Current reasoning ability.** Raven's Progressive Matrices (RPM) was employed for the present purpose to tap into current reasoning ability. RPM is a non-verbal test of intellectual functioning consisting of analogy problems in which a matrix of geometric figures is presented with one entry missing, and the correct missing entry must be selected from a set of answer choices. It is made of 60 multiple choice questions, listed in descending order of difficulty. Within the background of one's age, the performance of this measure has been reported in this manuscript in terms of percentile [34]. Thus, the total correct score out of 60 is calculated into percentile scores according to age using previously established normative data for Omanis [35].

**Cognitive Failure Questionnaire (CFQ).** The Cognitive Failure Questionnaire was used to assess various indices of cognitive failure [36]. CFQ has 25 items that tap into cognitive failures that have a direct bearing on daily life such as perception ('absent-mindedness'; "Do you fail to notice signposts on the road?"), memory ('memory deficit'; "Do you read something and find you haven't been thinking about it and must read it again?") and motor lapses ('slips of action'; "Do you bump into people?"). The items are answered on a 5-point Likert scale (0 = " never", 4 = "very often"). The summation of scores yields numbers ranging from 0–100. A higher score is often equated with cognitive decline. After obtaining approval from one of the progenitors of this scale (Katharine Parkes, Department of Experimental Psychology, University of Oxford), the Arabic version was generated using the protocol detailed elsewhere [37]. The internal consistency of this scale was > 0.85.

**Neuropsychological measures.** *Digit Span*. The Digit Span test (Wechsler Adult Intelligence Scale sub-test) was employed to tap into working memory, attention, and concentration. The scale has two components- Digits Forward and Digits Backward [38]. In the present context, only Digits Backward, in which the participant was read a random series of numbers and asked to repeat them in the reverse order (e.g., 9-1-3, should be repeated as 3-1-9), was employed. On this scale, lower scores denote the presence of cognitive decline.

*Buschke selective reminding test*. The Buschke Selective Reminding Test [39] taps into one's capacity to learn and remember. The participants are verbally presented with a 12-item list of unrelated words. Upon being read to them, participants are asked to recall as many items as they can remember, in any order, from the list of 12 items. After each recall trial, the examiner repeats only those items that the subject had forgotten during recall. The total items recalled out of the three trials are recorded as an outcome measure.

*Verbal fluency*. In the Controlled Oral Word Association Test, or Verbal Fluency [40], the participant is asked to verbally improvise as many different words as possible that begins with each of the three specific letters relevant to Arabic Speakers as described elsewhere [41]. On this scale, lower scores denote a higher propensity to neuropsychological dysfunction. Functional Magnetic Resonance Imaging (fMRI), positron emission tomography (PET) scan, and Single-photon emission computed tomography (SPECT) are often employed to tap into brain activity or cerebral blood flow associated with performance on neuropsychological tests. Using such functional brain scanning tools detects the association between verbal fluency and increased blood flow in the brain region, which is, in turn, often associated with executive

functioning, including the dorsolateral prefrontal cortex and its reciprocal projections in other brain regions [42].

*Trail Making Test (TMT)*. TMT has two versions [43]. For the present study, we have employed TMT-B. According to this, the participants are required to draw a line starting with a number and then a letter. Thus, he or she is required to shift between numbers (1–13) and letters (A-L) respectively. The total numbers and letters are drawn across are recorded in seconds. In this scale, a longer time taken denotes a problem with flexibility, divided attention, and working memory [44]. Zakzanis et al., have reported that greater brain activity in the dorsolateral and medial frontal was associated with performance on the TMT [45].

*Digit symbol*. Digit Symbol or Digit Symbol Substitution Test is one of the subscales in the Wechsler Adult Intelligence Scale [46], which taps into processing speed. Digit Symbol measures visual-motor speed. Scores may be affected by visual memory, coordination, and the ability to learn nonverbal material. The participant is shown numbers (1–9) accompanied by nine symbols (e.g., O, V, ⊥, ㄴ). In the blank space provided, the participants are required to write the correct numeral for the symbol below each symbol. A higher score is indicative of better cognitive performance, and conversely, lower scores denote the possible presence of cognitive decline [43].

**Affective ranges.** The Arabic version of the Hospital Anxiety and Depression Scale (HADS) was used to screen for the presence of anxiety and depressive symptoms. The scale has 14 items, of which seven are used for screening anxiety, and the other seven items are for depression. For both subscales, a score $\geq 8$ denotes 'case-ness' for either anxiety or depression [47].

## Statistical analysis

Data were analyzed by SPSS 23.0 (IBM SPSS Inc. Chicago, IL, USA), and the results of those endorsed work-life imbalance or work-life balance were analyzed using descriptive statistics. Univariate analysis was used, and demographics, current reasoning ability, and neuropsychological variables were evaluated with the chi-square test and independent t-test to reveal the association/difference between the work-life imbalance and balance groups. The research team then ran a multiple logistic regression analysis, where the work-life status is the dependent variable, and those variables with $p<0.05$ in the univariate analysis were the independent variables. This analysis could address the research aim to identify the contributing variables associated with work-life imbalance.

## Ethical approval and consent to participate

This study was approved by the Institutional Review Board -Medical Ethics Committee, College of Medicine and Health Science, Sultan Qaboos University (REF. NO. SQ U-EC/107/19, MREC Approval# 1913). Participants were requested to provide written informed consent and were carried out by the Code of Ethics of the World Medical Association (Declaration of Helsinki) for human experiments.

## Results

Table 1 shows the results for demographic, reasoning ability, and neuropsychological variables and their association with the status in work-life balance using a univariate and multivariate approach. A total of 168 subjects (75.3% of the responders) were considered to be at a work-life imbalance. The total sample had more women (53.4%) than men (46.6%) at an average age of 32.7 (SD = 11.3 years) years. Close to half of the sample (44.4%—n = 99) had a university

**Table 1. Descriptive statistics of study sample.**

| Variables/domains measured | | Total (n = 223) | |
|---|---|---|---|
| | | n (%) | Mean ± SD |
| *Demographic* | | | |
| Gender | Men | 104 (46.6) | |
| | Women | 119 (53.4) | |
| Education | Below University | 124 (55.6) | |
| | University | 99 (44.4) | |
| Income meets my needs | Just adequate/sufficient | 169 (75.8) | |
| | Not adequate/insufficient | 54 (24.2) | |
| Age | | | 32.7 ± 11.3 |
| *Current Reasoning Ability* | | | |
| *Raven Progressive Matrices* | Nonverbal Intelligence | | 44.9 ± 9.1 |
| *Ecological valid measure of cognitive decline* | | | |
| *Cognitive Failure Questionnaire* | Memory, absent-mindedness, and slips of action | | 37.2 ± 18.8 |
| *Neuropsychological measures* | | | |
| *Digit Span-backward* | Attention, and concentration | | 5.6 ± 2.5 |
| *Buschke Selective Reminding Test* | Learning and remembering | | 17.9 ± 6.2 |
| *Verbal fluency* | Executive Functioning | | 16.2 ± 4.9 |
| *Trail Making Test (Trial B)* | Executive Functioning | | 111.7 ± 58.6 |
| *Digit Symbol* | Processing Speed | | 63.2 ± 15.6 |
| *Affective Ranges* | | | |
| *Hospital Anxiety and Depression Scale (**Anxiety**)* | Yes | 185 (83.0) | 11.0 ± 4.0 |
| | No | 38 (17.0) | |
| *Hospital Anxiety and Depression Scale (**Depression**)* | Yes | 148 (66.4) | 8.6 ± 3.0 |
| | No | 75 (33.6) | |

education, and more than 75% (n = 169) reported that their income was just adequate/sufficient to meet their needs.

In the demographic variables, no significant associations were found between work-life status and gender (OR = 0.656, $p$ = 0.176), educational level (OR = 1.722, $p$ = 0.081), and income (OR = 0.626, $p$ = 0.229). Age was an exception ($t$ = 4.118, $p<0.001$). In the "Current" reasoning ability variables, a significant difference was found in work-life status on the subject's cognitive failure scores ($t$ = 4.146, $p < .001$) but not on their IQ percentile score ($t$ = 1.106, $p$ = 0.27). In the neuropsychological variables, a significant difference was found in most scores including work-life status on digit span-backward scores ($t$ = 11.693, $p < .001$), Buschke Selective Reminding Test scores ($t$ = 9.459, $p < .001$), verbal fluency score ($t$ = 11.854, $p < .001$), Trial Making scores ($t$ = 9.256, $p < .001$), and anxiety scores ($t$ = 3.368, $p$ = 0.001). The difference across the digit symbol scores ($t$ = 0.081, $p$ = 0.936) and depression scores ($t$ = 1.730, $p$ = 0.085) were not significant. A total of 185 (83.0%), and 148 (66.4%) subjects reported having anxiety and depression, respectively. For those who had anxiety, 137 subjects (74.1%) reported a work-life imbalance. Among those who endorsed depression, 114 subjects (77.0%) reported work-life imbalance.

In Table 2, multivariate analysis shows that demographic and neuropsychological variables were significantly associated with work-life imbalance. The model had a good-fit according to the Hosmer-Lemeshow goodness-of-fit test ($\chi^2$ = 0.333, $p$ = 0.998; Cox & Snell R square = 0.724), with a predicting power of 97.8%. Subjects who were older in age were more likely (Beta = -0.236, $p$ = 0.021) to have work-life imbalance compared to younger subjects.

**Table 2. Univariate and multivariate (multiple logistic regression) analysis for work-life imbalance in association with demographic, reasoning ability, and neuropsychological measurements.**

| | | Work-life | | Univariate analysis | | Multivariate analysis^ | |
|---|---|---|---|---|---|---|---|
| | | Imbalance (n = 168) | Balance (n = 55) | | | | |
| Variables | | n (%) | n (%) | Statistics | p-value | Beta | p-value |
| *Demographic* | | | | | | | |
| | Gender | | | | | | |
| | Men | 74 (44.0) | 30 (54.5) | 0.656[a] | 0.176 | | |
| | Women | 94 (56.0) | 25 (45.5) | | | | |
| | Education | | | | | | |
| | Below University | 99 (58.9) | 25 (45.5) | 1.722[a] | 0.081 | | |
| | University | 69 (41.1) | 30 (54.5) | | | | |
| | Income meets my needs | | | | | | |
| | Just adequate/sufficient | 124 (73.8) | 45 (81.8) | 0.626[a] | 0.229 | | |
| | Not adequate/insufficient | 44 (26.2) | 10 (18.2) | | | | |
| | Age | | | | | | |
| | Mean ± SD | 34.2 ± 11.8 | 28.2 ± 8.5 | 4.118[b] | < .001 | -0.236 | 0.021 |
| *Reasoning ability* | | | | | | | |
| | Raven Progressive Matrices | | | | | | |
| | Mean ± SD | 45.2 ± 10.3 | 44.2 ± 3.4 | 1.106[b] | 0.27 | | |
| | Cognitive Failure Scores | | | | | | |
| | Mean ± SD | 39.7 ± 19.3 | 29.5 ± 14.6 | 4.146[b] | < .001 | | |
| *Neuropsychological measures* | | | | | | | |
| | Digit Span-backward | | | | | | |
| | Mean ± SD | 4.9 ± 2.2 | 8.0 ± 1.5 | 11.693[b] | < .001 | 1.282 | 0.010 |
| | Buschke Selective Reminding Test | | | | | | |
| | Mean ± SD | 16.0 ± 5.2 | 23.7 ± 5.3 | 9.459[b] | < .001 | 0.415 | 0.007 |
| | Verbal fluency | | | | | | |
| | Mean ± SD | 14.2 ± 3.3 | 22.0 ± 4.5 | 11.854[b] | < .001 | 0.816 | 0.004 |
| | Trail Making Test Trial B | | | | | | |
| | Mean ± SD | 123.4 ± 62.9 | 75.9 ± 12.2 | 9.256[b] | < .001 | -0.264 | 0.001 |
| | Digit Symbol | | | | | | |
| | Mean ± SD | 63.2 ± 16.0 | 63.4 ± 14.5 | 0.081[b] | 0.936 | | |
| | Anxiety | | | | | | |
| | Yes | 137 (81.5) | 48 (87.3) | 0.644[a] | 0.327 | | |
| | No | 31 (18.5) | 7 (12.7) | | | | |
| | Mean ± SD | 11.4 ± 4.4 | 9.8 ± 2.2 | 3.368[b] | 0.001 | -0.343 | 0.048 |
| | Depression[c] | | | | | | |
| | Yes | 114 (67.9) | 34 (61.8) | 1.304[a] | 0.411 | | |
| | No | 54 (32.1) | 21 (38.2) | | | | |
| | Mean ± SD | 8.8 ± 2.9 | 8.0 ± 3.0 | 1.730[b] | 0.085 | | |

a, $\chi^2$ test, Odds Ratio;

b, Independent t-test, |t|-statistics;

c, Yes (score 8–21) and No (score < 8).

^, Backward stepwise (Wald); Hosmer and Lemeshow Test, $\chi^2$ = 0.333, p = 0.998; Cox & Snell R Square = 0.724; Sensitivity = 99.4%, Specificity = 92.7%, overall predicting power = 97.8%.

Subjects who reported impaired ranges in the *Trail Making Test* (Beta = -0.264, *p* = 0.001) and fell within case-ness for anxiety (Beta = -0.343, *p* = 0.048) were more likely to indicate a work-life imbalance. Participants who reported scores in the impaired ranges in the *Digit Span* (Beta = 1.282, *p* = 0.010), *Buschke Selective Reminding* (Beta = 0.415, *p* = 0.007), and Verbal Fluency (Beta = 0.816, *p* = 0.004) tests were more likely to report a work-life imbalance.

## Discussion

Various paradigms have been used to explore predicaments of those in occupational settings including seminal work on occupational stress and burnout [48]. There is also complementary data on the types and magnitude of psychological distress found in organizational settings [49]. More recently, work-place balance and cognition have been increasingly reported in the context of occupational settings. To our knowledge, work-place balance and cognition have recently scant attention and nonwestern society is no exception.

This study sought to explore the subjective judgment of work-life imbalance among individuals referred to a tertiary care center in Oman. A total of 168 participants (75.3%) judged themselves to have a work-life imbalance. In previous surveys, the rate was 10.8% in the Netherlands [50], 30% in Switzerland [51], and 23.2% in the USA [7]. The comparison of the magnitude of work-life dissatisfaction is hampered by the fact that the majority of the studies were conducted among special populations. For example, healthcare professionals and those who work in shifts [32, 52–55]. Shift workers, by definition, are known to have impacted circadian rhythm, which in itself is likely to compromise health and cognitive functioning [29, 30]. The present study obtained information from the cadre of employees in the different caliber of jobs. Those are participants who are under scrutiny for fitness to work. Therefore, they constitute a convenient and selective group. In support of this view, the present participants were those who were referred by their place of employment for psychometric evaluation in order to determine fitness to work and if they are permitted for sick leave and/or receiving a less taxing workload. Such factors leading for referral could have some implications for the data and results. For example, the data illustrates that more participants have indicated they have work-life imbalance than those who have work-life balance. This could be due to some confounding factor, such as malingering in order to get sick leave or have a less taxing workload. Other confounding factors could include a self-fulfilling prophecy where participants were referred because of some work performance issues so the participants may be more willing to state they have work-life imbalance. One way to circumvent these confounders is to recruit those group of participants who were not referred for the psychometric evaluation as comparative group. Extension to this that future studies should extend to the general working population.

Despite such a caveat, to our knowledge, this is the first study from the Arabian Gulf population, which, in addition to exploring the presence of work-life balance, has also examined whether work-life balance is associated with specific neuropsychological functioning and mood states. In the psychiatric population, cognitive symptoms have been shown to persist despite remission of other psychological symptoms [56, 57]. Similarly, Oosterholt et al. have reported that after ten weeks of psychological intervention, individuals with burnout showed no improvement in cognition [23]. The authors postulated that burnout tends to lead to 'permanent cognitive deficits. While the merit of such a theory would require further scrutiny, it nevertheless suggests the importance of exploring neuropsychological functioning in the occupational setting.

It is interesting to note that the two groups did not differ in socio-demographic variables except for age. The participants who were older age were more likely to have work-life imbalance than their younger counterparts. This finding concurs with other studies suggesting that

the natural course of stress and burnout follows an accumulative effect [58]. From an organizational perspective, early identification of individuals with work-life imbalance would support early mitigation and would prevent decreased productivity, burnout, and eventually turnover. Letting the imbalance drag overtime would only exacerbate the detrimental effects on the efficiency of the organization and the satisfaction of customers. Policy and decision-makers are encouraged by the findings to establish programs that would ensure early identification of, and intervention against, work-life imbalance.

It is also worthwhile to note that performance on indices of intellectual capacity, *Raven's Progressive Matrices*, did not differ between the two groups. Since IQ did not differ between the two groups and considering that the decline in IQ is likely to affect cognition [59], findings suggest that the observed neuropsychological differences are independent of IQ. This finding underscores the pivotal role played by the work culture and context, which, if unhealthy, could precipitate the loss of productivity and eventually turnover of intellectually competent individuals. This realization has led many countries to dedicate attention and resources to the creation of healthy and supportive work environments to enhance the productivity and longevity of human capital [60]. On the positive side, the findings promise a positive return on investment for supportive programs and work arrangements aiming at helping employees regain their work-life balance since such individuals, once they regain their balance, would have the intellectual ability and environmental knowledge to swiftly contribute to the organization.

Published literature shows that most work-life balance interventions involved the provision of employee benefits and services such as paid parental leave, job reorganizing such as flexible work hours, and initiatives to embed work-life balance within organizational cultures [61]. In the United States, Family and Medical Leave Act (FMLA) has been recognized as a work-life benefit that helps employees balance their work and family demands by permitting unpaid but job-secure leave from work for their own or their families' needs [62]. This leave has been associated with increased job satisfaction, productivity, and retention, in addition to reduced work-family conflicts [62]. A study by Panda & Sahoo revealed the importance of human resource interventions, specifically team building and communication for maintaining a proper balance between work life and family life [63]. Another study by Michel, Bosch, & Rexroth [64] among German employees showed that practicing mindfulness is associated with better psychological detachment from work, less strain-based work-family conflict, and greater work-life balance. Similarly, there is evidence to suggest that mindfulness techniques have been associated with neuroplasticity [65]. It remains to be seen whether neuroplasticity associated with mindfulness has the potential to heighten the integrity of neuropsychological functioning.

This study utilized five neuropsychology measures tapping into attention and concentration, learning and remembering, executive functioning and processing speed. While significant differences were observed in scores on indices of attention and concentration (*Digit Span)*, learning and remembering *(Buschke Selective Reminding Test)*, executive functioning (*Verbal Fluency, Trail Making Test)*, and the processing speed, as tapped into by Digit Symbol test, was not significantly different. The results suggest that processing speed may not be an appropriate test to use when examining the neuropsychological effects of work-life imbalance. More importantly, the findings underscore the detrimental effects that the work-life imbalance has on serious and essential elements of job performance. While the indices associated with work-life imbalance would affect all sectors of employment, they will be most impactful in jobs requiring focus and concentration and those jobs whose outcomes are serious or irreversible (e.g. healthcare). Particular attention needs to be dedicated to the investigation of work-life balance in sectors where lack of concentration and poor attention may lead to catastrophic consequences. Future studies could also use longitudinal designs to understand whether the impaired neuropsychologic functions are the cause or the effect of work-life imbalance.

While the two groups did significantly differ in objective batteries of attention and concentration, learning and remembering, and executive functioning, they did not differ in self-reported cognitive difficulties. Previous studies have suggested high concordance between conventional neuropsychology batteries and indices of memory, absent-mindedness, or slips of action as tapped into by the *Cognitive Failure Questionnaire* [66]. But there is a dissenting view in the findings of this study [67]. Socio-Cultural patterns might contribute to the noted discrepancy in endorsing self-reported cognitive difficulties [68]. While recall bias might be a factor, in Oman and some other cultures from the East, what is known as introspective is generally not patterned in society [69]. Individuals growing up in Oman are raised with the values of a collectivistic society. Future studies should explore whether socio-cultural factors play a role in how one answers self-reported measures.

Mood state or affective range was tapped into by this study using the *Hospital Anxiety and Depression Scale*. Literature is replete with studies suggesting the 'epidemic' of anxiety and depression among people at the workplace [70–72]. Some studies have shown afflictive mood state has a strong relationship with burnout [73]. For those whose scores indicated case-ness on indices of anxiety, 74.1% reported an imbalance in life-work. Similarly, depressive symptoms were endorsed by about 77% of the participants with a work-life imbalance. The significant association between work-life balance and mental illness is disconcerting and supports the calls for a national program to screen for work-life imbalance and intervene with programs and services to help employees regain such balance once an imbalance is detected. A prerequisite of such programs would be the development of reliable measures for work-life balance and the political resolve to institute flexible and supportive employment policies and practices that are often missing in the Middle East Region.

## Limitations

The study has several limitations. Firstly, despite the abundance of studies on work-life balance, there are no proper psychometrically established measures that categorize whether one is 'balanced' in work-life or otherwise. Based on the present literature, on one hand, the use of a work-life instrument in the current study appears to have a clear cut-off point between different grades of work-life balance. On the other hand, the presence of work-life balance or imbalance was solicited by two dimensions of the questionnaire that led into two grouping—those with work-life balance vs. work-life imbalance. Studies are needed to dissect the concept of work-life balance further. There are likely to be gender and cultural differences in the perception of work-life balance, which would also require an empirical examination. Secondly, the present study was not equipped to differentiate between whether the observed cognitive functioning constitutes impairment or otherwise. Future studies should examine whether the presence of observed performance constitutes impairment that would warrant contemplation of relevant prevention or rehabilitation. Third, it is not clear whether the observed differences might stem from pre-existing issues that were not explored in this study. Non-communicable diseases, sometimes called lifestyle diseases, are rife in Oman [74]. It is conceivable that pre-existing conditions that would have a direct bearing on the integrity of cognitive functioning are present in some of the participants. Future studies should have more robust mechanisms in place to rule out such confounders. Related to this, future studies should decipher the type of occupation of the participants. Fourth, the cross-sectional design of this study did support the establishment of causality, yet it allowed the identification of significant associations that could be investigated further in the future. Fifth, the target population were individuals referred for assessment due to poor performance at work and thus the external validity of the findings of this study is limited to similar individuals and may not be representative of the general population.

## Conclusion

Work-life imbalance has been shown to impact not only work motivation and performance, but also the spectrum of emotional and cognitive functioning. In this study, we explored whether Omanis referred for fitness to work evaluation at different stages of work-life balance show impaired general information processing, cognition or specifically neuropsychological functioning. Compared to those maintaining a healthy work-life balance, those reporting work-life imbalance differ significantly on indices of attention and concentration, learning and remembering and executive functioning. The participants also differed significantly scores on indices of affective ranges with those with work life imbalance endorsed high rate of anxiety. The current preliminary study from non-western population suggests heuristic value of bringing neuropsychological tools into come to grip with predicament at the organizational setting. If the present finding will withstand further scrutiny, the implication of impaired neuropsychological functioning should be contemplated, as well as finding mechanism to designing valid work-life balance measures, systematic assessment of occupational burnout and the continuous evidence-based improvement of the quality of practice environment. It is essential for policy and decision makers at the system (regulators, ministries) and the institutional levels (managers and leaders) assess the extent of work-life balance for employees and intervene with support programs and initiative to help individuals regain the balance in their life. Such interventions would be necessary to enhance productivity and decrease absenteeism and turnover in the workforce.

## Acknowledgments

The authors would like to thank the participants for their time and effort to undertake protracted cognitive and intellectual assessments.

## Author Contributions

**Conceptualization:** Samir Al-Adawi, Muna Al-Saadoon, Amal A. Al Balushi, Hamad Al-Kindy, Saud M. Al-Harthi.

**Data curation:** Samir Al-Adawi, Muna Al-Saadoon, Amal A. Al Balushi, Hamad Al-Kindy, Saud M. Al-Harthi.

**Formal analysis:** Samir Al-Adawi, Amal A. Al Balushi, Moon Fai Chan.

**Funding acquisition:** Samir Al-Adawi.

**Writing – original draft:** Samir Al-Adawi, Mohamad Alameddine, Moon Fai Chan, Karen Bou-Karroum.

**Writing – review & editing:** Samir Al-Adawi, Mohamad Alameddine, Moon Fai Chan, Karen Bou-Karroum.

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
