## [Decision Letter · Decision Letter 0]

23 Dec 2021

PONE-D-21-19931

The magnitude and effect of work-life imbalance on cognition and affective range among the non-western population: A study from Muscat

PLOS ONE

Dear Dr. Samir Al-Adawi,

Thank you for submitting your manuscript to PLOS ONE. After careful consideration, we feel that it has merit but does not fully meet PLOS ONE’s publication criteria as it currently stands. Therefore, we invite you to submit a revised version of the manuscript that addresses the points raised during the review process.

We look forward to receiving your revised manuscript.

Kind regards,

Rogis Baker, Ph.D

Academic Editor

PLOS ONE

Journal Requirements:

2. PLOS ONE does not copy edit accepted manuscripts (https://journals.plos.org/plosone/s/criteria-for-publication#loc-5). To that effect, please ensure that your submission is free of typos and grammatical errors. 

*Please change "female” or "male" to "woman” or "man" as appropriate, when used as a noun (see for instance https://apastyle.apa.org/style-grammar-guidelines/bias-free-language/gender). 

4. We noticed you have some minor occurrence of overlapping text with the following previous publication(s), which needs to be addressed:

-https://www.sjweh.fi/show_abstract.php?abstract_id=3256

The text that needs to be addressed involves the entirety of the abstract. 

In your revision ensure you cite all your sources (including your own works), and quote or rephrase any duplicated text outside the methods section. Further consideration is dependent on these concerns being addressed.

Reviewers' comments:

Reviewer's Responses to Questions

**Comments to the Author**

1. Is the manuscript technically sound, and do the data support the conclusions?

Reviewer #1: Partly

Reviewer #2: Yes

2. Has the statistical analysis been performed appropriately and rigorously? 

Reviewer #1: Yes

Reviewer #2: Yes

3. Have the authors made all data underlying the findings in their manuscript fully available?

Reviewer #1: No

Reviewer #2: Yes

4. Is the manuscript presented in an intelligible fashion and written in standard English?

Reviewer #1: Yes

Reviewer #2: Yes

5. Review Comments to the Author

Reviewer #1: The manuscript aims to examine work-life balance/imbalance. The manuscript used a series of questionnaires and neuropsychological test batteries in order to assess the relationship between cognitive functioning and work-life imbalance in a sample of participants referred for an evaluation by their work. The results of the experiment suggest that age, self-reported cognitive failure, cognitive test batteries, and anxiety predict work-life imbalance. That is, those who are older, have worse cognitive functioning scores, and more anxiety are more likely to report having work-life imbalance.

Overall, the conducted research in the manuscript aims to fill in the gap in the research involving the relationship between cognitive functioning and work-life imbalance across individuals from varying occupations. The manuscript is sufficient and concise. However, there are some concerns regarding rational for the hypotheses, results of the data, and interpretations.

Major Concerns:

1. The manuscript begins by discussing the relationship between burnout and poor coping at the workplace, but it is unclear what this relationship is. In addition, the manuscript suggests that it aims to investigate if burnout leads to work-life imbalance or if work-life imbalance leads to burnout (lines 67-69) but this is not reflected in the hypothesis or data.

a. In line with this, the manuscript suggests that there is a relationship between job burnout and work-life imbalance but there is limited research on how this may affect or be affected by cognitive functioning. There is a wealth of research on burnout and cognitive functioning that should be addressed.

2. It is unclear which aspect of cognitive functioning the manuscript is interested in studying. Cognitive functioning is a big umbrella term and should be defined to the reader in addition to discussing which aspect of cognitive functioning is most affected by burnout or work-life imbalance (i.e., working memory, long-term memory, cognitive control, etc…).

3. The hypothesis suggests that some sort of group comparison between work-life balance vs. work-life imbalance will be made in the statistical analysis but the statistics reflecting a group comparison appears to be missing.

a. In line with this, the regression analysis appears to be only conducted for the work-life imbalance group, which does not support the group comparison suggested in the hypothesis.

b. Table 2 of the manuscript reports data regarding the mean of each group (balance vs. imbalance) for different outcome measures. The table also reports separate univariate regression and multivariate regression coefficients. It is unclear whether the manuscript ran separate univariate regression analyses for each variable and a multivariate analysis for other variables. It is also unclear which variables were controlled for in the multivariate analysis. The primary interest of this manuscript was the relationship between cognition and work-life imbalance and data regarding anxiety and depression was included. Were these mood related variables controlled for in the analysis? If the goal of the manuscript is to compare the balance vs. the imbalance groups, consider the manuscript may want to alternative or additional analyses.

4. Hypothesis 3 of the manuscript states that it aims to examine the relationship between work-life balance and symptoms of depression and anxiety. However, this aim was not reviewed in the introduction.

5. Lines 110-119: The manuscript stated that the participants recruited for the present study were those who were referred by their place of employment for a psychological evaluation in order to determine fitness to work and if they are permitted for sick leave and/or receiving a less taxing workload. The explanation provided for the reason for this referral could have some implications for the data and results. For example, the data illustrates that more participants have indicated they have work-life imbalance than those who have work-life balance. This could be due to some confounding factor, such as malingering in order to get sick leave or have a less taxing workload. Other confounding factors could include a self-fulfilling prophecy where participants were referred because of some work performance issues so the participants may be more willing to state they have job burnout. Motivation may play a key factor. There are many studies on motivation and cognitive functioning and motivation and burnout that should be considered as alternative explanations for the results of the manuscript.

a. In line with this point, the manuscript mentioned that participants who were referred to the psychological evaluation completed a work-life questionnaire where answers on two dimensions of the questionnaire determined their grouping in the balance vs. imbalance. Could it be that these participants felt job burnout or work-life imbalance and as a reason their work performance suffered, and they were sent this referral? Data collection from a group of participants who were not referred for the psychological evaluation could provide further clarity.

Minor Concerns:

1. Lines 66-67: The manuscript introduces many terms and not all are defined to the reader. Please consider only using the term/construct that is relevant to the goal of the manuscript.

2. Lines 70-71: Does the manuscript mean imbalance rather than balance?

3. Line 142: The manuscript describes how all measures were scored expect for the RPM. For consistency, consider providing this information for the RPM.

4. Line 141: It is unclear what aspect of cognition the CFQ is assessing (attention, LTM, WM…).

Reviewer #2: It's clear and interesting. There are not enough papers with this subject on this population. In the abstract you use the term "mood" without explanation and definition and you never use this term in next paragraph.

6. PLOS authors have the option to publish the peer review history of their article (what does this mean?). If published, this will include your full peer review and any attached files.

Reviewer #1: **Yes: **Lilian Azer

Reviewer #2: **Yes: **Ophelie Bouillet

---

## [Author Response · Author response to Decision Letter 0]

14 Jan 2022

Response to Reviewers: point-counterpoint

Manuscript Title: The magnitude and effect of work-life imbalance on cognition and affective range among the non-western population: A study from Muscat

Manuscript ID: PONE-D-21-19931

Review Comments to the Author Authors' response to Reviewers

Reviewer #1 (Lilian Azer): The manuscript aims to examine work-life balance/imbalance. The manuscript used a series of questionnaires and neuropsychological test batteries in order to assess the relationship between cognitive functioning and work-life imbalance in a sample of participants referred for an evaluation by their work. The results of the experiment suggest that age, self-reported cognitive failure, cognitive test batteries, and anxiety predict work-life imbalance. That is, those who are older, have worse cognitive functioning scores, and more anxiety are more likely to report having work-life imbalance.

Overall, the conducted research in the manuscript aims to fill in the gap in the research involving the relationship between cognitive functioning and work-life imbalance across individuals from varying occupations. The manuscript is sufficient and concise. However, there are some concerns regarding rational for the hypotheses, results of the data, and interpretations. 

Thank you for your positive feedback and the helpful comments. This revised version of the manuscript was modified in line with the comments and suggestions.

Major Concerns: 1. The manuscript begins by discussing the relationship between burnout and poor coping at the workplace, but it is unclear what this relationship is. In addition, the manuscript suggests that it aims to investigate if burnout leads to work-life imbalance or if work-life imbalance leads to burnout (lines 67-69) but this is not reflected in the hypothesis or data.

a. In line with this, the manuscript suggests that there is a relationship between job burnout and work-life imbalance but there is limited research on how this may affect or be affected by cognitive functioning. There is a wealth of research on burnout and cognitive functioning that should be addressed. 

We are happy to provide additional clarification on this point. The context of bringing burnout and poor coping is to lay the groundwork that the focus has been largely on burnout and poor coping (poor mental health outcome etc). This study did not investigate if burnout leads to work-life imbalance or if work-life imbalance leads to burnout (lines 67-69). Rather, as we stated in the text, the aims have been to (i) explore the rate of work-life imbalance among Omanis with poor adjustment at work, (ii) compare demographic, reasoning ability, and neuropsychological factors between people has indicated work-life balance and work-life imbalance, and (iii) explore the factors that contributing to the work-life imbalance. We have added a statement on the relationship between poor coping and burnout, please check page 4 in the main manuscript. 

As for the second point (“There is a wealth of research on burnout and cognitive functioning that should be addressed”), we agree with this statement. However, our focus is on variation among work-life balance and cognition and affective ranges (anxiety and depression). 

 2. It is unclear which aspect of cognitive functioning the manuscript is interested in studying. Cognitive functioning is a big umbrella term and should be defined to the reader in addition to discussing which aspect of cognitive functioning is most affected by burnout or work-life imbalance (i.e., working memory, long-term memory, cognitive control, etc…). 

Thank you for this helpful comment.

In the methods section, further details on the domain for each of the cognitive measures that tap into has now been added. Similarly, this issue was also narrated in the introduction. As it is widely known, most of these measures are attributed to different cognitive domains. For example, the presently employed digit span has been linked to working memory, attentional span, executive function. However, in the manuscript, we have operationalized them and used them consistently.

 3. The hypothesis suggests that some sort of group comparison between work-life balance vs. work-life imbalance will be made in the statistical analysis but the statistics reflecting a group comparison appears to be missing. 

Thank you for pointing this out. The statistics we used to compare work-life imbalance and work-life balance were indicated in Table 2 under the univariable analysis. For example, those people who indicated work-life imbalance were older (34.2 years) than work-life balance (28.2 years, t=4.118, p<.001). The result is to address the aim (ii) of this study. Please refer to the Introduction, lines 111-113.

 a. In line with this, the regression analysis appears to be only conducted for the work-life imbalance group, which does not support the group comparison suggested in the hypothesis. 

Thank you for this comment. The logistic regression explores factors associated with work-life imbalance. It is to address the study aim (iii). Please refer to the Introduction, lines 111-113.

 b. Table 2 of the manuscript reports data regarding the mean of each group (balance vs. imbalance) for different outcome measures. The table also reports separate univariate regression and multivariate regression coefficients. It is unclear whether the manuscript ran separate univariate regression analyses for each variable and a multivariate analysis for other variables. It is also unclear which variables were controlled for in the multivariate analysis. The primary interest of this manuscript was the relationship between cognition and work-life imbalance and data regarding anxiety and depression was included. Were these mood related variables controlled for in the analysis? If the goal of the manuscript is to compare the balance vs. the imbalance groups, consider the manuscript may want to alternative or additional analyses. 

Thank you for this comment. To explore factors associated with work-life imbalance, we first used t-test or chi-square to examine any sig. difference between work-life imbalance and work-life balance groups on each variable (univariate analysis). Those variables that were sig. <0.05 in the univariate analysis were put in the logistic regression (multivariate analysis) to explore factors contributing to the work-life imbalance. All details are shown in the statistical analysis and Table 2.

 4. Hypothesis 3 of the manuscript states that it aims to examine the relationship between work-life balance and symptoms of depression and anxiety. However, this aim was not reviewed in the introduction. 

Thank you for raising this point. We agree that this issue needs to be reviewed in the introduction. Now a paragraph has been inserted, please check page 5 in the main manuscript.

 5. Lines 110-119: The manuscript stated that the participants recruited for the present study were those who were referred by their place of employment for a psychological evaluation in order to determine fitness to work and if they are permitted for sick leave and/or receiving a less taxing workload. The explanation provided for the reason for this referral could have some implications for the data and results. For example, the data illustrates that more participants have indicated they have work-life imbalance than those who have work-life balance. This could be due to some confounding factor, such as malingering in order to get sick leave or have a less taxing workload. Other confounding factors could include a self-fulfilling prophecy where participants were referred because of some work performance issues so the participants may be more willing to state they have job burnout. Motivation may play a key factor. There are many studies on motivation and cognitive functioning and motivation and burnout that should be considered as alternative explanations for the results of the manuscript. 

Thank you for raising this issue. Important indeed. These are important confounders that limit the generalization of our study. We have therefore taken them as limitations of the study, now narrated in the discussion. Please check pages 15-16 in the main manuscript. 

 a. In line with this point, the manuscript mentioned that participants who were referred to the psychological evaluation completed a work-life questionnaire where answers on two dimensions of the questionnaire determined their grouping in the balance vs. imbalance. Could it be that these participants felt job burnout or work-life imbalance and as a reason their work performance suffered, and they were sent this referral? Data collection from a group of participants who were not referred for the psychological evaluation could provide further clarity. 

These are valid confounder. These issues have been elaborated in the discussion/limitations sections. Please check pages 19-20 in the main manuscript.

Minor Concerns: 1. Lines 66-67: The manuscript introduces many terms and not all are defined to the reader. Please consider only using the term/construct that is relevant to the goal of the manuscript. 

Thank you for raising this point. The terms ‘work-life balance’, ‘work-family conflict’, ‘work-nonwork interface’ or its counterpart, ‘work-life imbalance’, ‘work-life conflict’ have now been defined in the text. The reason that we want to keep them is that the field is new and different terms are used despite that they all allude to the same concept. Please check page 3 in the main manuscript. 

 2. Lines 70-71: Does the manuscript mean imbalance rather than balance?

 Thank you for the opportunity to revise the manuscript and correct any typos. The word ‘imbalance’ is most appropriate. We have replaced it please check page 3 in the main manuscript.

 3. Line 142: The manuscript describes how all measures were scored expect for the RPM. For consistency, consider providing this information for the RPM.

 As suggested by the reviewer, the scoring for RPM is now narrated in the text. Please check page 7 in the main manuscript. 

 4. Line 141: It is unclear what aspect of cognition the CFQ is assessing (attention, LTM, WM…). 

We are happy to elaborate this point further. The domains CFQ measures now have been detailed in the newly added part of Table 1 as well in the description of the measure in the Method. Please check page 7 in the main manuscript. 

Reviewer #2

(Ophelie Bouillet) It's clear and interesting. There are not enough papers with this subject on this population. In the abstract you use the term "mood" without explanation and definition and you never use this term in next paragraph.

 Thank you for these kind words and for pointing out the abrupt entrance of the term ‘mood’. The proper terms have now been narrated, that is, we have employed indices of anxiety and depressive symptoms. Please check page 2 in the main manuscript.

---

## [Editor Report · Decision Letter 1]

24 Jan 2022

The magnitude and effect of work-life imbalance on cognition and affective range among the non-western population: A study from Muscat

PONE-D-21-19931R1

Dear Dr. Samir Al-Adawi,

We’re pleased to inform you that your manuscript has been judged scientifically suitable for publication and will be formally accepted for publication once it meets all outstanding technical requirements.

Kind regards,

Rogis Baker, Ph.D

Academic Editor

PLOS ONE
---

## [Editor Report · Acceptance letter]

26 Jan 2022

PONE-D-21-19931R1 

The magnitude and effect of work-life imbalance on cognition and affective range among the non-western population: A study from Muscat 

Dear Dr. Al-Adawi:

I'm pleased to inform you that your manuscript has been deemed suitable for publication in PLOS ONE. Congratulations! Your manuscript is now with our production department. 

Kind regards, 

on behalf of

Dr. Rogis Baker 

Academic Editor

PLOS ONE